# Genome-wide association study of leprosy in Malawi and Mali

**James J. Gilchrist** [1,2,3]*, **Kathryn Auckland**[3], **Tom Parks**[3,4], **Alexander J. Mentzer**[3], **Lily Goldblatt**[5], **Vivek Naranbhai**[3], **Gavin Band**[3], **Kirk A. Rockett**[3], **Ousmane B. Toure**[6], **Salimata Konate**[6], **Sibiri Sissoko**[6], **Abdoulaye A. Djimdé**[6], **Mahamadou A. Thera**[6], **Ogobara K. Doumbo**[6†], **Samba Sow**[7], **Sian Floyd**[8], **Jörg M. Pönnighaus**[9], **David K. Warndorff**[9], **Amelia C. Crampin**[9], **Paul E. M. Fine**[8], **Benjamin P. Fairfax**[2], **Adrian V. S. Hill**[3,10]*

**1** Department of Paediatrics, University of Oxford, Oxford, United Kingdom, **2** MRC–Weatherall Institute of Molecular Medicine, University of Oxford, Oxford, United Kingdom, **3** Wellcome Centre for Human Genetics, University of Oxford, Oxford, United Kingdom, **4** Department of Infectious Diseases, Imperial College London, London, United Kingdom, **5** Balliol College, Oxford, United Kingdom, **6** Malaria Research and Training Centre, University of Science, Techniques and Technologies of Bamako, Bamako, Mali, **7** Center for Vaccine Development, Bamako, Mali, **8** Faculty of Epidemiology and Population Health, London School of Hygiene & Tropical Medicine, London, United Kingdom, **9** Malawi Epidemiology and Intervention Research Unit (formerly Karonga Prevention Study), Chilumba, Malawi, **10** Jenner Institute, University of Oxford, Oxford, United Kingdom

† Deceased.

* james.gilchrist@paediatrics.ox.ac.uk (JJG); adrian.hill@ndm.ox.ac.uk (AVSH)

**Data Availability Statement:** GWAS summary statistics are available through the NHGRI-EBI GWAS Catalog (https://www.ebi.ac.uk/gwas/downloads/summary-statistics; accession codes: Malawi, GCST90129399; Mali, GCST90129400; meta-analysis, GCST90129401). Code and source

## Abstract

Leprosy is a chronic infection of the skin and peripheral nerves caused by *Mycobacterium leprae*. Despite recent improvements in disease control, leprosy remains an important cause of infectious disability globally. Large-scale genetic association studies in Chinese, Vietnamese and Indian populations have identified over 30 susceptibility loci for leprosy. There is a significant burden of leprosy in Africa, however it is uncertain whether the findings of published genetic association studies are generalizable to African populations. To address this, we conducted a genome-wide association study (GWAS) of leprosy in Malawian (327 cases, 436 controls) and Malian (247 cases, 368 controls) individuals. In that analysis, we replicated four risk loci previously reported in China, Vietnam and India; MHC Class I and II, *LACC1* and *SLC29A3*. We further identified a novel leprosy susceptibility locus at 10q24 (rs2015583; combined $p = 8.81 \times 10^{-9}$; $OR = 0.51$ [95% CI 0.40 − 0.64]). Using publicly-available data we characterise regulatory activity at this locus, identifying *ACTR1A* as a candidate mediator of leprosy risk. This locus shows evidence of recent positive selection and demonstrates pleiotropy with established risk loci for inflammatory bowel disease and childhood-onset asthma. A shared genetic architecture for leprosy and inflammatory bowel disease has been previously described. We expand on this, strengthening the hypothesis that selection pressure driven by leprosy has shaped the evolution of autoimmune and atopic disease in modern populations. More broadly, our data highlights the importance of defining the genetic architecture of disease across genetically diverse populations, and that disease insights derived from GWAS in one population may not translate to all affected populations.

data underlying the findings of the study are available here: https://doi.org/10.5281/zenodo.7130158. The terms of consent under which MalariaGEN participants were recruited require managed access to individual level genotype data. Genotype and phenotype data describing the MalariaGEN Malian samples have been deposited with the European Genome-Phenome Archive (accession code EGAD00010001737). Data is accessible by application to an independent data access committee, details of which are available here: https://ega-archive.org/datasets/EGAD00010001737.

**Funding:** JJG and AJM are funded by National Institute for Health Research (NIHR - https://www.nihr.ac.uk) Clinical Lectureships. During this work AVSH was supported by a Wellcome Trust (https://wellcome.org) Senior Investigator Award (HCUZZ0) and by a European Research Council (https://erc.europa.eu) advanced grant (294557). The research was supported by the Wellcome Trust Core Award Grant Number 203141/Z/16/Z with additional support from the NIHR Oxford BRC (https://oxfordbrc.nihr.ac.uk). The views expressed are those of the author(s) and not necessarily those of the NHS, the NIHR or the Department of Health and Social Care. The funders had no role in study design, data collection and analysis, decision to publish, or preparation of the manuscript.

**Competing interests:** The authors have declared that no competing interests exist.

## Author summary

Leprosy remains a leading cause of infectious disability globally. Human genetic variation is a major determinant of susceptibility to infection, including leprosy. Large-scale genetic association studies have been pivotal in advancing our understanding of leprosy biology. These studies have been performed in Chinese, Vietnamese and Indian populations, and it remains unclear whether these insights are informative of leprosy susceptibility in African populations. To address this, we performed a genome-wide association study of leprosy susceptibility in Malawi and Mali. In doing so we replicate known leprosy susceptibility loci at MHC class I and II, *LACC1* and *SLC29A3*. Furthermore, we identify a novel leprosy susceptibility locus, which has been under recent selection pressure, which demonstrates pleiotropy with inflammatory bowel disease and atopic disease. We identify expression of *ACTR1A* in $CD4^+$ T cells as candidate effector of leprosy risk at this locus. These data deepen our understanding of leprosy biology and further implicate this ancient pathogen in the evolution of immune-mediated diseases in modern populations.

## Introduction

Leprosy is a chronic infectious disease affecting the skin and peripheral nerves caused by *Mycobacterium leprae*. It is a leading infectious cause of disability [1]. The introduction of multidrug therapy [2], widespread use of BCG vaccination [3], and the 1991 World Health Assembly resolution to eliminate leprosy by the year 2000 have all contributed to a decline in disease burden; nevertheless, over 200,000 new cases continue to be reported annually (https://www.who.int/gho/neglected_diseases/leprosy/en/), numbers which are likely to represent a considerable underestimate of the true disease burden [4].

Large-scale, unbiased genetic association studies in Chinese [5–10], Indian [11] and Vietnamese [12] populations have identified and validated 34 genetic loci independently associated with leprosy outside the HLA region, as well as independent HLA class I and class II associations. A key feature of these studies has been the demonstration of considerable genetic heterogeneity in leprosy susceptibility between populations. For instance, while genetic variation at *TLR1* associates with leprosy risk in Indian and Turkish populations, this finding has not been replicated in Chinese and Vietnamese populations. To date, there are no published genome-wide association studies (GWAS) of leprosy in African populations. This is important as there remains a significant burden of leprosy in Africa and the observed inter-population heterogeneity reported in the Chinese, Vietnamese and Indian studies suggest that the published GWAS findings may not be generalizable to African populations. To address this, we have performed a GWAS of leprosy susceptibility in Malawian and Malian individuals.

## Results

### Leprosy genome-wide association study

Individuals with leprosy were recruited to the study following clinical and microbiological evaluation within the Karonga Prevention Study (KPS), Karonga, Malawi. Healthy controls were recruited from the same population. Following quality control and genome-wide imputation, genotypes at 10,511,695 loci from 612 samples (284 cases, 328 controls) were included in the association analysis (Table 1). Inspection of the QQ plot (S1 Fig) and the genomic inflation parameter ($\lambda = 1.0333$) demonstrates that inclusion of the six major principal components as covariates in the model adequately controls for confounding variation. In that analysis, we

**Table 1. Demographics & clinical characteristics of study samples.**

| | | Numbers (% total) | Sex (female) | Age (months) | Genotyping platform |
|---|---|---|---|---|---|
| Malawi cases | Overall | 284 | 172 (61%) | 45 (15–83) | Illumina ADPC |
| | PB | 242 (88%) | 153 (63%) | 57 (32–72) | |
| | MB | 34 (12%) | 15 (44%) | 42 (15–83) | |
| Malawi controls | | 328 | 206 (63%) | 42 (15–82) | Illumina ADPC |
| Mali cases | Overall | 208 | 62 (30%) | 46 (13–85) | Illumina ADPC |
| | PB | 67 (32%) | 45 (67%) | 45 (13–80) | |
| | MB | 141 (68%) | 39 (28%) | 47 (17–85) | |
| | ENL | 107 (38%)* | | | |
| | RR | 42 (15%)* | | | |
| Mali controls (Set 1) | | 142 | 91 (64%) | 30 (14–67) | Illumina ADPC |
| Mali controls (Set 2) | | 169 | 84 (50%) | 3 (0–15) | Illumina Omni 2.5M |

*These numbers represent cases recruited to the original study, n = 279, of which the current study samples are a subset. Leprosy reaction data is not available for Malawi case samples.

PB, paucibacillary; MB, multibacillary; ENL, erythema nodosum leprosum; RR, reversal reaction; ADPC, African Diaspora Power Chip.

identified 142 loci, representing 38 independent genomic loci, with suggestive evidence of association ($p < 1 \times 10^{-5}$) with leprosy in Malawian populations (S2 Fig, S1 Table).

## GWAS replication and meta-analysis

We sought to replicate evidence for leprosy association observed in Malawi among leprosy cases and healthy controls in Mali. Individuals with leprosy were recruited to the study at Mali's former national leprology centre, Institut Marchoux, (now Hôpital Dermatologique de Bamako), Bamako. Healthy controls were recruited from the same population. Among the Malian samples, 10,514,676 loci and 519 individuals (208 cases, 311 controls) passed QC filters (Table 1). The QQ plot (S1 Fig) and genomic inflation parameter ($\lambda = 1.0498$) of genome-wide association analysis in the Malian samples demonstrates control of confounding variation with inclusion of the major six major principal components and genotyping platform as covariates in the model. We combined evidence for leprosy association in Malawi and Mali using a fixed-effects meta-analysis (Fig 1). Of the 142 leprosy-associated loci identified in the discovery analysis, 18 SNPs, at a single genomic locus at 10q24.32 (Fig 2A), showed evidence of replication in Mali ($p < 0.05$) and overall evidence of association with leprosy exceeding genome-wide significance ($p < 5 \times 10^{-8}$). The variant with the strongest evidence for leprosy association at that locus is rs2015583 (Table 2): $p = 8.81 \times 10^{-9}$, OR = 0.51 (95% CI 0.40−0.64). There is no evidence for heterogeneity of effect between populations at rs2015583 (heterogeneity $p = 0.871$, Fig 2B), and the data best supports a model in which rs2015583 modifies risk of both paucibacillary and multibacillary leprosy (log10 Bayes factor = 6.01, Fig 2B). Fine-mapping of the leprosy association at chr10q24.32 identifies a credible set of 32 SNPs with a 99% probability of containing the causal variant, spanning a 53kb region: chr10:104,225,316–104,278,276 (S3 Table). Genetic variation at this locus has not been previously described as a determinant of leprosy risk.

To understand whether genotype at rs2015583 could act as a determinant of mycobacterial disease in Africa more broadly, we examined the role of rs2015583 in genetic association studies of tuberculosis in Ghana (1,359 cases, 1,952 controls) and The Gambia (1,316 cases, 1,382 controls). In that analysis, genotype at rs2015583 is not associated with tuberculosis in Ghana

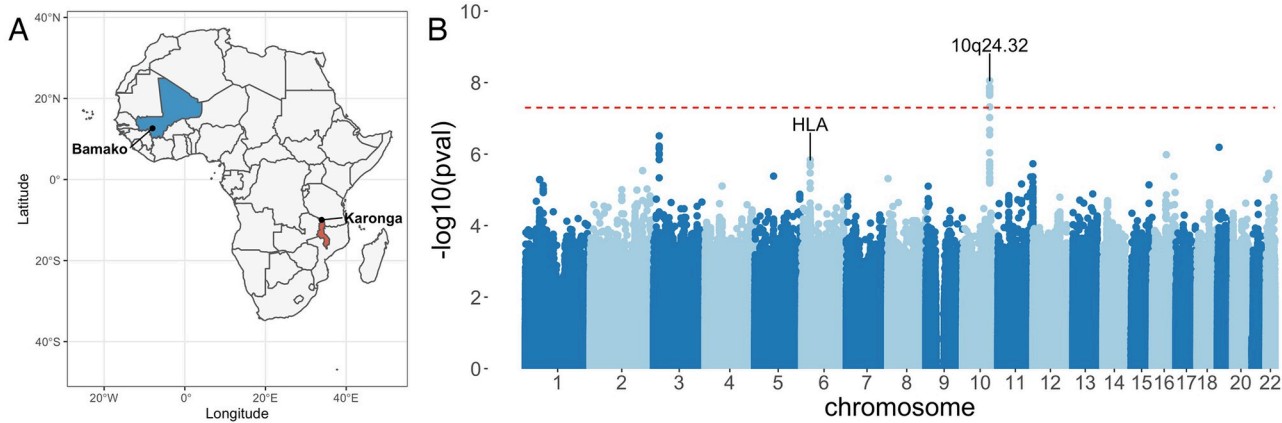

**Fig 1. Genome-wide association study of leprosy in Malawi and Mali.** (A) Study sites in Mali (blue) and Malawi (red). (B) Evidence for association with leprosy at genotyped and imputed autosomal SNPs and indels ($n = 9, 616, 523$) in Malawi and Mali (492 cases, 639 controls). Association statistics represent a fixed-effects meta-analysis of additive association with disease in Malawi and Mali. The red, dashed line denotes genome-wide significance ($p = 5 \times 10^{-8}$). Suggestive loci ($p = 1 \times 10^{-5}$) identified in the meta-analysis are detailed in S2 Table. The map is based on public domain Natural Earth data; the base layer is available for download at https://www.naturalearthdata.com/.

($p = 0.746$, $OR = 1.02$ (95% CI $0.92 - 1.03$)) or The Gambia ($p = 0.112$, $OR = 0.91$ (95% CI $0.81 - 1.02$)).

## Leprosy risk at rs2015583 in non-African populations

To explore whether genetic variation at chromosome 10q24.32 is a determinant of leprosy susceptibility in other settings, we examined evidence for leprosy association at rs2015583 in Chinese and Indian populations using previously-published datasets describing the genetics of leprosy risk [5, 8, 9, 11]. We obtained summary statistics of leprosy association at rs2015583 generated as part of six genetic association studies in four Chinese populations [5, 8, 9]. There is no evidence for leprosy association in any individual Chinese population tested (S4 Table, minimum $p = 0.171$) or in a fixed-effects meta-analysis of all six genotyping experiments (2,743 cases, 3,573 controls) at rs2015583 (Fig 2C, $p = 0.587$, $OR = 0.98$ (95% CI $0.90 - 1.06$)).

To explore the role of genotype at rs2015583 in leprosy risk in India, we used previously-published genotyping data from a leprosy case-control collection of Indian-ancestry individuals recruited in New Delhi [11]. In 448 individuals (209 cases and 239 controls), genotype at rs2015583 is not associated with risk of leprosy overall ($p = 0.299$, $OR = 0.80$ (95% CI $0.52 - 1.22$)). However, in a multinomial regression model, in which cases are stratified by disease subtype, genotype at rs2015583 is associated with risk of multibacillary disease (Fig 2C, $p = 0.044$, $OR = 0.56$ (95% CI $0.32 - 0.98$)) but not paucibacillary disease ($p = 0.692$, $OR = 1.11$ (95% CI $0.66 - 1.86$)).

## The leprosy risk locus has pleiotropic effects on risk of immune-mediated disease

A key of feature of genetic determinants of leprosy described to date has been the identification of pleiotropy at leprosy risk loci with other immune-mediated diseases, in particular inflammatory bowel disease [13, 14]. To explore whether leprosy-associated genetic variation identified in our study is a determinant of other immune-mediated diseases in human populations, we assessed evidence for colocalization of the leprosy association at chr10q24.32 with immune-mediated diseases (n = 29) and haematological traits (n = 26). In that analysis (Fig

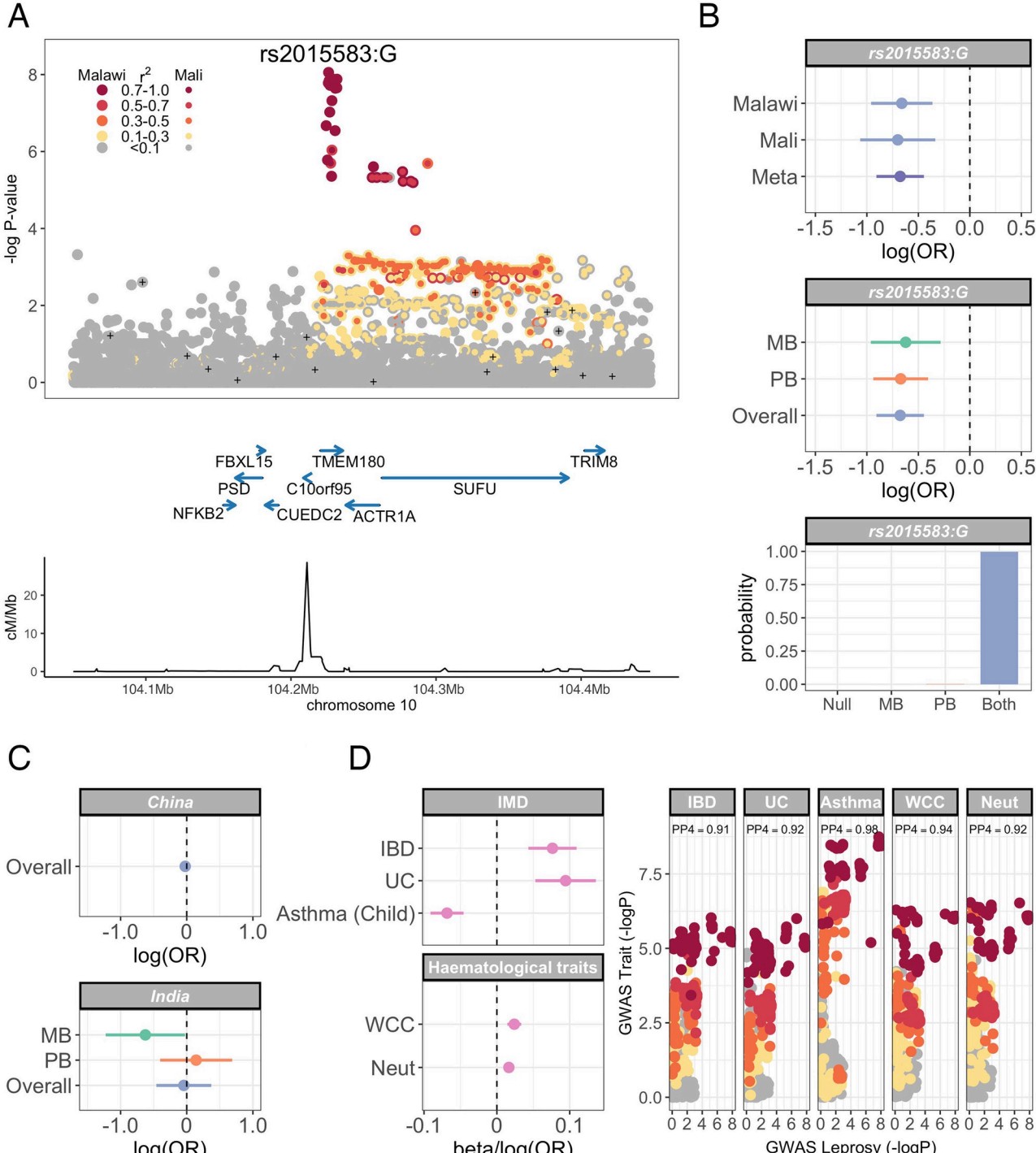

**Fig 2. Leprosy association and pleiotropy at chromosome 10q24.32.** (A) Regional association plot of leprosy association at chr10q24.32. Association statistics represent a fixed-effects meta-analysis of additive association with disease in Malawi and Mali. SNPs are coloured according to linkage disequilibrium to rs2015583, and genotyped SNPs marked with black plusses. (B) Log-transformed odds ratios and 95% confidence intervals of rs2015583 association with leprosy in Malawi and Mali (top) and stratified by multibacillary and paucibacillary disease (middle). Posterior probabilities of models of rs2015583 association with leprosy: "Null", no association with leprosy; "MB", non-zero effect in multibacillary leprosy alone; "PB", non-zero effect in paucibacillary leprosy alone; "Both", the same non-zero effect is shared by individuals with multibacillary and paucibacillary leprosy. (C) Log-transformed odds ratios and 95% confidence intervals of rs2015583 association with leprosy in China (top) and in India (bottom). Effect estimates in China represent risk of leprosy overall. Effect estimates in India are stratified by multibacillary and paucibacillary disease. (D) Log-transformed odds ratios and 95% confidence intervals (left) of rs2015583 association with immune-mediated diseases (IBD, inflammatory bowel disease; UC, ulcerative

colitis; childhood-onset asthma) and hematological indices (WCC, white cell count; Neut, neutrophil count). The leprosy association signal colocalizes with the GWAS locus for each trait at chr10q24.32 (right). SNPs are coloured according to linkage disequilibrium to rs2015583 as above.

2D, S5 Table), the leprosy association colocalizes with three immune-mediated diseases (inflammatory bowel disease, ulcerative colitis and childhood-onset asthma), and two haematological traits (white cell count and neutrophil count). Carriage of the G allele at rs2015583 is associated with decreased risk of leprosy and childhood-onset asthma, but increased risk of inflammatory bowel disease and increased neutrophil and white cell counts.

## Signatures of selection at the leprosy-associated locus

The identification of a pleiotropic locus for leprosy, autoimmune and atopic disease risk suggests a hypothesis in which selection pressure imposed by leprosy has shaped the evolution of immune-mediated disease in modern populations. To further explore this we evaluated evidence for recent positive selection at the identified locus, examining integrated haplotype scores (iHS) [15] in 1000 Genome Project populations. In that analysis, we find evidence of positive selection (rank $p < 0.05$) at a genomic region (chr10:104270877–104280877), which overlaps with the leprosy-associated locus, in 16 of 25 1000 Genome Project populations (S6 Table), with the strongest evidence of selection (iHS>2) observed in 4 African ancestry populations (ASW, LWK, GWD and MSL).

## Regulatory function at the leprosy risk locus

Trait-associated genetic variation identified by GWAS are highly enriched for regulatory variation. To understand whether the identified leprosy-associated variation at chromosome 10q24.32 was likely to operate through regulation of gene expression we explored overlap between SNPs in the 99% credible set (n = 32) and functional chromatin states in nine cell types using ENCODE annotations (S7 Table) [16]. Overlap of credible set SNPs with regulatory chromatin states is most marked in K562 cells (a myeloid leukaemia cell line), with 11 of 32 credible set SNPs overlapping with chromatin states suggestive of regulatory function; 7 with strong enhancers, 3 with weak enhancers and one with an active promoter. One 99% credible set SNP, rs2274351:T, lies within an active promoter in all nine cell types (chr10:104,260,000–104,265,601), which is located at the 5' end of two genes; *ACTR1A* and *SUFU*. Moreover, rs2274351 is located within the binding motifs of two transcription factors; CNOT3 and TRIM28.

To further explore whether leprosy-associated genetic variation at chr10q24.32 modifies leprosy risk through its effect on gene expression, we investigated whether the leprosy risk locus colocalises with expression quantitative trait loci (eQTL) in skin, peripheral nerves and a

**Table 2. Effect of rs2015583 genotype on risk of leprosy in Malawi & Mali.**

|  |  | Numbers | Genotypes | MAF | OR (95% CI) | p-value |
|---|---|---|---|---|---|---|
| Malawi | Cases | 284 | 11/90/183 | 0.196 | 0.52 (0.38–0.7) | $p = 6.99 \times 10^{-6}$ |
|  | Controls | 328 | 27/137/164 | 0.292 |  |  |
| Mali | Cases | 208 | 12/71/125 | 0.229 | 0.50 (0.34–0.72) | $p = 1.14 \times 10^{-4}$ |
|  | Controls | 311 | 55/151/105 | 0.418 |  |  |
| Meta-analysis | Cases | 492 | 23/137/308 | 0.21 | 0.51 (0.4–0.64) | $p = 8.81 \times 10^{-9}$ |
|  | Controls | 639 | 82/288/269 | 0.354 |  |  |

MAF, minor allele frequency. CI, confidence interval.

range of primary immune cells; monocytes [17], B cells [18], NK cells [19], neutrophils [20], CD4$^+$ T cells and CD8$^+$ T cells [21]. In keeping with the observation that a credible set SNP for the leprosy association overlaps with an active promoter 5' to *ACTR1A*, among eQTL mapping data in primary immune cells (Fig 3A), there is evidence for colocalisation between the leprosy risk locus and an eQTL for *ACTR1A* expression in CD4$^+$ T cells (posterior probability of colocalisation, *PP4* = 0.94). We observe no evidence for colocalisation between the leprosy risk locus and gene expression in other primary immune cells (Fig 3A, S8 Table). The leprosy protective allele, rs2015583:G, is associated with reduced *ACTR1A* expression in CD4$^+$ T cells ($p = 4.69 \times 10^{-9}$, $\beta = -0.085$). In skin and peripheral nervous tissue, however, there is evidence

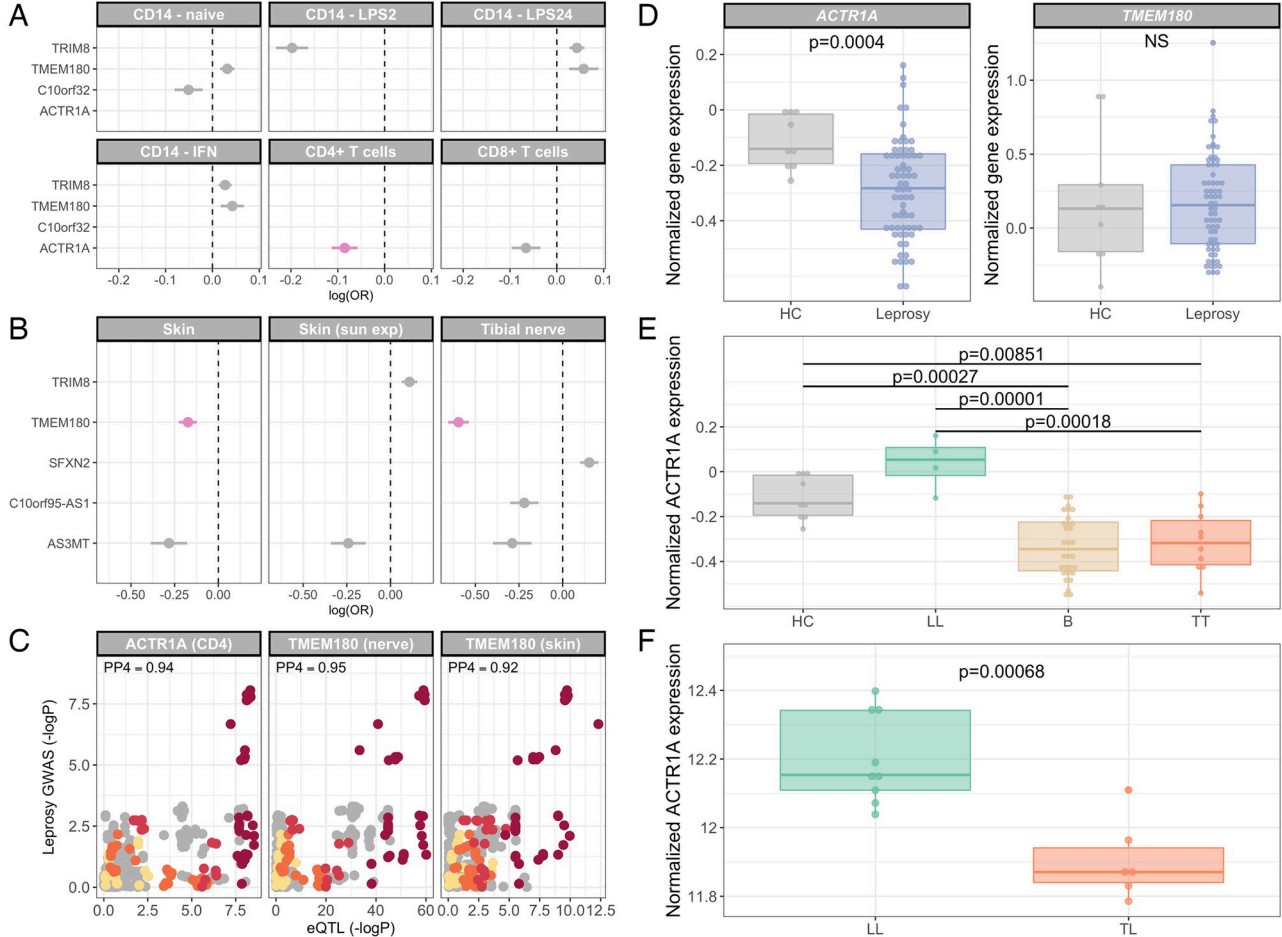

**Fig 3. ACTR1A and TMEM180 are candidate mediators of leprosy susceptibility at chromosome 10q24.32.** (A) Association of rs2015583:G genotype with gene expression in primary immune cells. Associations for which there is evidence of a shared causal variant with the leprosy association are highlighted (pink). (B) Association of rs2015583:G genotype with gene expression in skin (sun exposed and unexposed) and peripheral nervous tissue (tibial nerve). Associations for which there is evidence of a shared causal variant with the leprosy association are highlighted (pink). (C) Evidence for colocalisation of the leprosy association signal at chr10q24.32 with *ACTR1A* expression in CD4$^+$ T cells and *TMEM180* expression in skin (sun unexposed) and peripheral nerves. SNPs are coloured according to linkage disequilibrium to rs2015583 as above. PP4, posterior probability of signal colocalisation. (D) *ACTR1A* and *TMEM180* expression in skin biopsy samples from healthy controls (HC, n = 9) and skin lesions from leprosy patients (n = 67) from the Belone *et al* microarray data [22]. (E) *ACTR1A* expression in skin biopsy samples from healthy controls (HC, n = 9) and skin lesions from leprosy patients with lepromatous (LL, n = 4), tuberculoid (TT, n = 10) and borderline (B, n = 30) disease from the Belone *et al* microarray data [22]. (F) *ACTR1A* expression in skin biopsy samples from leprosy patients with lepromatous (LL, n = 9) and tuberculoid (TT, n = 6) disease from the Montoya *et al* dataset [23]. P-values for comparison of two groups were derived with t-tests (normally distributed data) or Mann-Whitney tests (non-normally distributed data). Comparison of expression across multiple groups was performed by ANOVA, with subsequent pairwise testing with Tukey's HSD tests.

for colocalisation between the leprosy risk locus and an eQTL for *TMEM180* expression (Fig 3B, *PP4* = 0.92 and *PP4* = 0.95 respectively), with the leprosy protective allele being associated with reduced expression of *TMEM180* in both cell types (skin—sun unexposed, $p = 1.84 \times 10^{-10}$, $\beta = -0.174$; tibial nerve, $p = 1.07 \times 10^{-59}$, $\beta = -0.596$).

## Candidate gene expression in leprosy

Having identified two distinct candidates in *ACTR1A* and *TMEM180* for mediation of the leprosy risk effect at chromosome 10q24.32, we sought to explore the expression of both genes in the context of leprosy. To do this we downloaded publicly-available gene expression data in whole blood and skin from patients with leprosy and healthy controls [22–25], exploring evidence for differential expression of *ACTR1A* and *TMEM180* in leprosy. There is no evidence of differential gene expression of *ACTR1A* ($p = 0.747$) or *TMEM180* ($p = 0.069$) in the whole blood of household contacts of leprosy patients in Bangladesh who do not develop disease (n = 40) and those that do (n = 40), either before onset of disease or at diagnosis (S3 Fig) [25]. In keeping with this, genotype at rs2015583 is not associated ($p < 0.05$) with expression of any gene in *cis* in the whole blood of Vietnamese borderline leprosy patients (n = 51) either unstimulated or following stimulation with sonicated *M. leprae* (S4 Fig) [24].

We then explored whether expression *ACTR1A* or *TMEM180* was perturbed in leprosy skin lesions. In comparison to healthy skin samples (n = 9), gene expression of *ACTR1A* is significantly down-regulated in skin lesions (n = 67) from patients with leprosy in Brazil (Fig 3D, $p = 0.0004$) [22]. In these data there is no evidence for differential expression of *TMEM180* in leprosy skin lesions (Fig 3D, $p = 0.822$). To further explore the down-regulation of *ACTR1A* expression in leprosy skin lesions, we investigated the effect of subtype of clinical leprosy on *ACTR1A* expression in the same dataset [22]. In that analysis (Fig 3E) the down-regulation of *ACTR1A* expression in leprosy skin lesions compared to healthy controls is seen in tuberculoid ($p = 0.0085$) and borderline forms of leprosy ($p = 0.0003$), but not in lepromatous disease ($p = 0.218$). We replicated this finding in a complementary RNA-Seq dataset of leprosy skin lesions from individuals with lepromatous leprosy (n = 9) and tuberculoid disease (n = 6) [23]. In these data we again see significant down-regulation of *ACTR1A* expression in lesions of leprosy patients with tuberculoid disease compared to lepromatous disease (Fig 3F, $p = 0.0007$).

## Replication of leprosy HLA associations

The observation that class I and II HLA alleles are key determinants of leprosy risk has been highly reproducible across diverse populations [8, 11, 12]. Motivated by this, and by evidence of association in the HLA observed in our data (Fig 1), we explored evidence for leprosy association in the HLA in Malawi and Mali at the level of SNPs and classical HLA alleles. In a fixed-effects meta-analysis of leprosy association in Malawi and Mali (Fig 4A, S9 Table), the peak classical allele association is a class II allele: HLA-DQB1*04:02 ($p = 6.74 \times 10^{-5}$, *FDR* = 0.0063, *OR* = 2.1 95% CI 1.75 − 2.51). We also observed a leprosy association in the class I region, at HLA-B*49:01 ($p \times 6.02 \times 10^{-4}$, *FDR* = 0.0156), which is independent of HLA-DQB1*04:02 (S5 Fig). No significant residual associations were observed after conditioning on both DQB1*04:02 and HLA-B*49:01 (S6 Fig). There is no evidence for heterogeneity of effect between populations at HLA-DQB1*04:02 or HLA-B*49:01 (heterogeneity $p = 0.8836$ and $p = 0.7049$, Fig 4B), and the data best supports a model in which both HLA-DQB1*04:02 and HLA-B*49:01 modify risk of both paucibacillary and multibacillary leprosy (log10 Bayes factors = 2.22 and 0.43 respectively, Fig 4C and 4D). HLA-DRB1*15:01 has been identified as a dominant determinant of leprosy risk in Chinese populations [8], with confirmatory evidence of association in Vietnamese [26] and Brazilian [27] populations. In our study samples

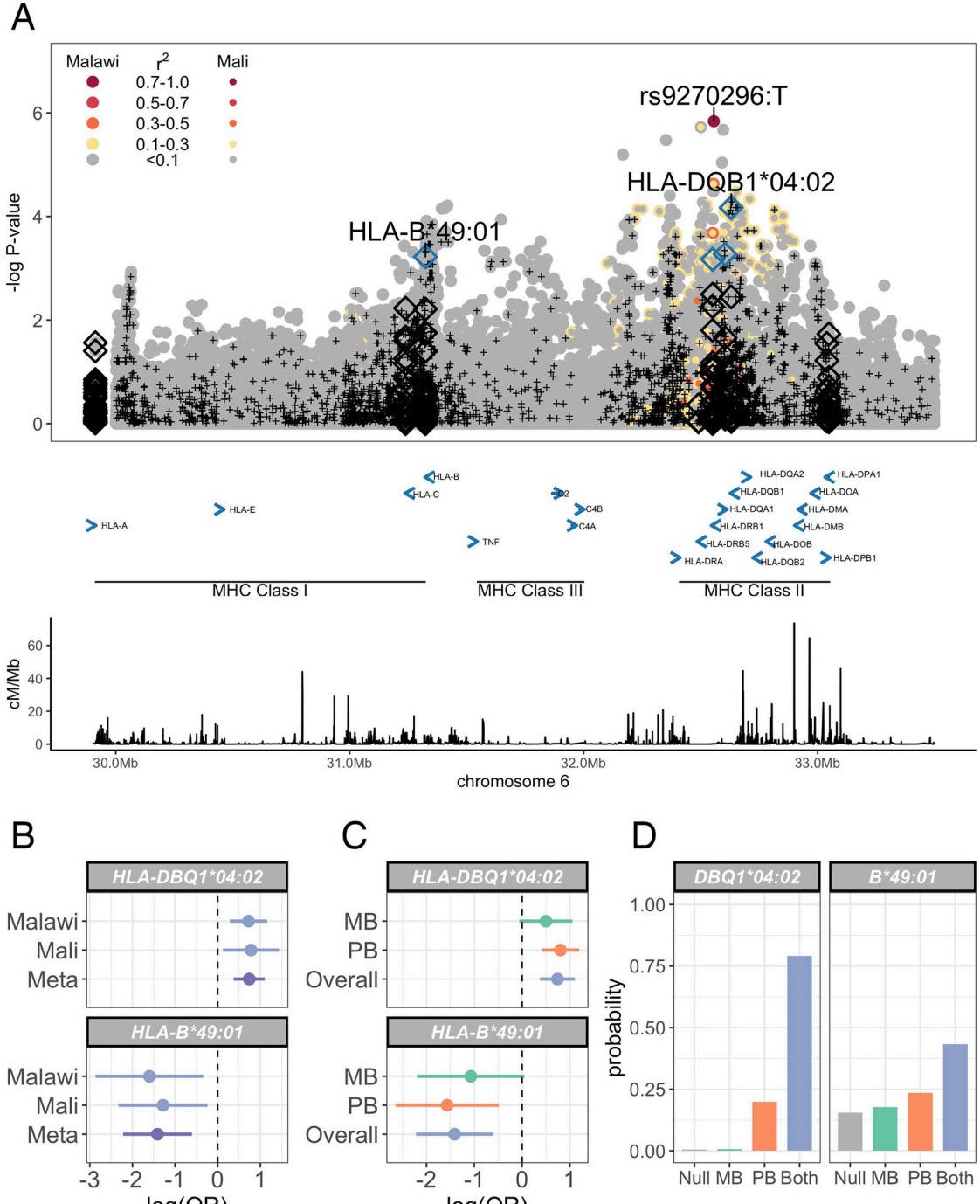

**Fig 4. MHC leprosy association in Malawi and Mali.** (A) Regional association plot of leprosy association across the HLA region. Association statistics represent a fixed-effects meta-analysis of additive association with disease in Malawi and Mali. SNPs are coloured according to linkage disequilibrium to rs9270926, and genotyped SNPs marked with black plusses. Imputed classical HLA alleles are plotted as diamonds, with significantly associated (FDR <0.05) alleles highlighted in blue. (B) Log-transformed odds ratios and 95% confidence intervals of HLA-DBQ1*04:02 and HLA-B*49:01 associations with leprosy in Malawi and Mali. (C) Log-transformed odds ratios and 95% confidence intervals of HLA-DBQ1*04:02 and HLA-B*49:01 associations with leprosy stratified by multibacillary and paucibacillary disease. (D) Posterior probabilities of models of HLA-DBQ1*04:02 and HLA-B*49:01 associations with leprosy: "Null", no association with leprosy; "MB", non-zero effect in multibacillary leprosy alone; "PB", non-zero effect in paucibacillary leprosy alone; "Both", the same non-zero effect is shared by individuals with multibacillary and paucibacillary leprosy.

HLA-DRB1*15:01 carriage is rare, with only 2 individuals carrying HLA-DRB1*15:01 alleles in Malawi (both cases) and no HLA-DRB1*15:01 alleles observed in Mali.

## Replication of known leprosy associations outside the HLA

Our identification of a genetic variant modifying leprosy risk in African populations, but not in Chinese populations, highlights the inter-population heterogeneity that has been observed across large-scale genetic studies of leprosy susceptibility [8, 11, 12]. We thus sought to replicate evidence for leprosy association at genetic loci outside the HLA associated with leprosy risk in other populations. In this analysis we included variants associated with leprosy susceptibility at genome-wide significance in other populations [5–11, 28–30], alongside variants at *LRRK2* [9] and *PRKN* [30], and pathogenic variants at *NOD2* [31]. We have previously published a replication analysis at rs3764147 (the leprosy-associated SNP at *LACC1*) in the Mali samples [32]. Of 37 leprosy risk loci assessed, 3 were not reliably imputed in our dataset (rs145562243, rs6871626, rs1873613) and 10 (rs146466242, rs925368, rs671, rs149308743, rs142179458, rs75680863, rs2066844, rs76418789, rs5743291, rs663743) were monomorphic or at very low frequency (MAF<1%) in our study samples (S10 Table). We were thus able to assess replication at 24 loci, at which our study had adequate power (>80%) to replicate ($p < 0.05$) findings at 7 loci. We were able to replicate leprosy associations at 2 loci (Fig 5A); a missense SNP in *LACC1*, rs3764147 ($p = 0.004$, $OR = 1.36$ 95% CI $1.10 − 1.67$), and a missense SNP in *SLC29A3*, rs780668 ($p = 0.034$, $OR = 1.28$ 95% CI $1.02 − 1.60$). There is no evidence for heterogeneity of effect between populations at rs3764147 or rs780668 (heterogeneity $p = 0.444$ and $p = 0.159$, Fig 5A), and the data best supports a model in which both rs3764147 and rs780668 modify risk of both paucibacillary and multibacillary leprosy (log10 Bayes factors = 1.64 and 0.68 respectively, Fig 5B and 5C).

Our failure to replicate evidence for leprosy association at more than 2 loci reflects a lack of study power at 17 of 24 SNPs assessed (S10 Table). Our study power is in part a reflection of our modest sample numbers, but is also contributed to by allele frequency differences between discovery populations and our study samples. At 20 of 37 loci, the allele frequency difference between that reported in the discovery analysis and our study samples was >10%, of which 14 resulted in decreased power to detect a genetic association in African populations (S10 Table). A striking example of this is at the *IL18RAP/IL18R1* locus (rs2058660), at which the discovery population MAF is 0.49 [29], falling to 0.10 and 0.11 in Malawi and Mali respectively. We further considered whether our failure to replicate previously-reported leprosy associations could represent differential linkage disequilibrium to an undefined causal locus between study populations. To test this, we examined evidence for leprosy association within 250kb of each previously-reported leprosy risk locus outside the HLA. In that analysis we identified a promoter variant in *RAB32*, rs34271799, with suggestive evidence of association with leprosy risk in Malawi and Mali ($p_{MALAWI} = 0.0023$, $p_{MALI} = 0.0034$, $p_{COMBINED} = 6.00 \times 10^{-5}$; $OR = 0.42$, 95% $CI = 0.28 − 0.64$, S7 Fig). There was no evidence suggestive of leprosy association ($p < 1 \times 10^{-4}$) within 250kb of any other previously identified leprosy risk locus. Finally, we assessed whether any failure of replication may represent differences in proportion of paucibacillary and multibacillary leprosy between our study samples and other sample collections. In that analysis there is no evidence that our lack of replication reflects effects restricted to multibacillary or paucibacillary disease (S11 Table).

## Discussion

In this study, we demonstrate that genetic variation at chromosome 10q24.32 is a determinant of leprosy risk in African populations. We expand upon this, identifying evidence of recent

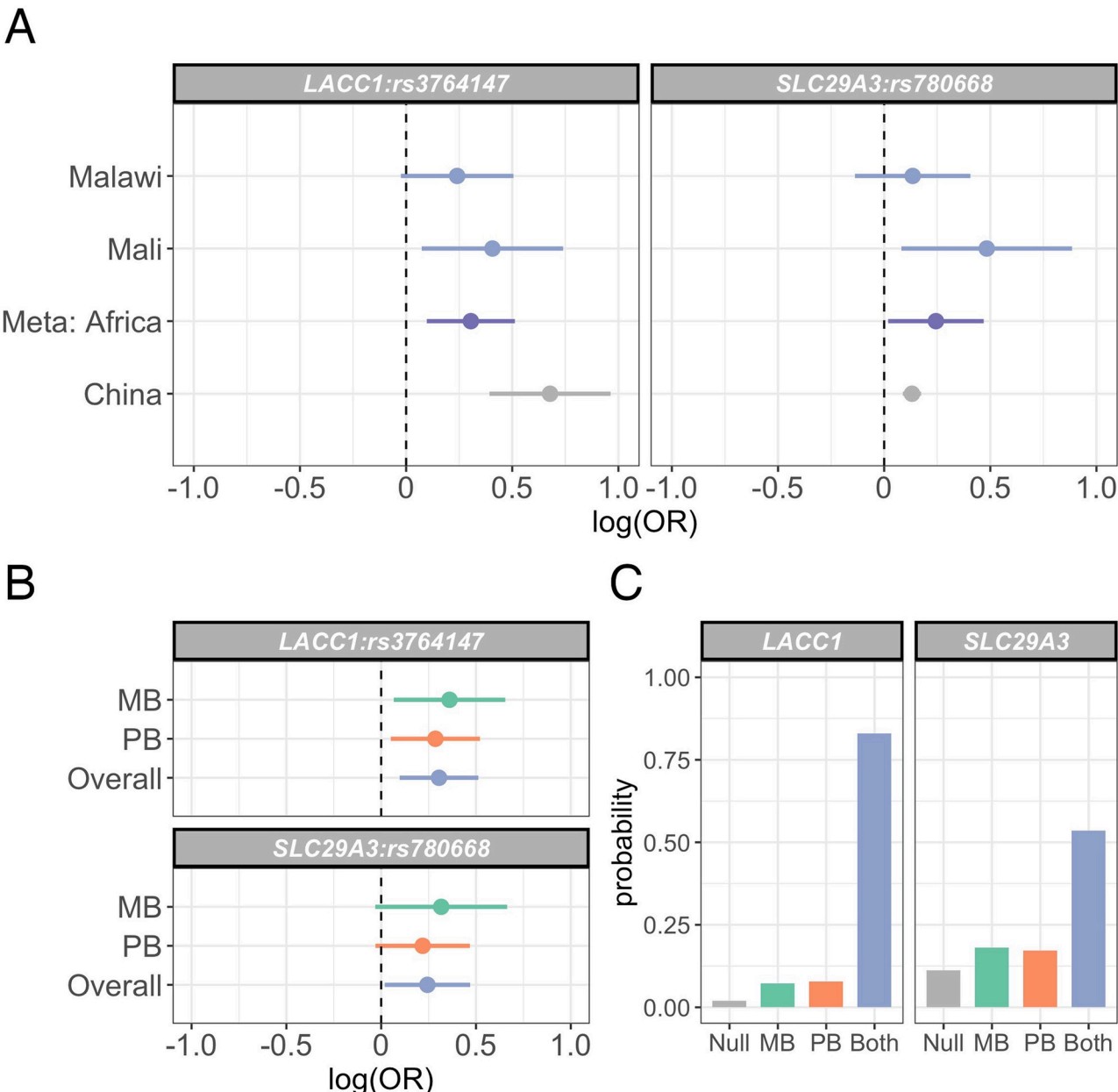

**Fig 5. Replication of leprosy associations at LACC1 and SLC29A3 in Malawi and Mali.** (A) Log-transformed odds ratios and 95% confidence intervals of rs3764147 and rs780668 associations with leprosy in Malawi and Mali. (B) Log-transformed odds ratios and 95% confidence intervals of rs3764147 and rs780668 associations with leprosy stratified by multibacillary and paucibacillary disease. (C) Posterior probabilities of models of rs3764147 and rs780668 associations with leprosy: "Null", no association with leprosy; "MB", non-zero effect in multibacillary leprosy alone; "PB", non-zero effect in paucibacillary leprosy alone; "Both", the same non-zero effect is shared by individuals with multibacillary and paucibacillary leprosy.

positive selection at this locus as well as evidence of pleiotropy, suggesting a shared genetic architecture of leprosy, inflammatory bowel disease and atopy. In common with many examples of trait-associated genetic variation identified by GWAS [33], there is evidence for regulatory activity at 10q24.32, and by integrating data describing chromatin state, eQTL mapping and differential gene expression in leprosy, we identify *ACTR1A* as a candidate mediator of

leprosy risk. Furthermore, we replicate previously identified leprosy susceptibility loci at *LACC1*, *SLC29A3*, and with HLA Class I and II alleles.

Our identification of a genetic locus modifying leprosy susceptibility in African populations, but with no effect on leprosy risk in well-powered GWAS in Chinese populations, is consistent with the existence of heterogeneity of genetic architecture of leprosy risk between populations. In keeping with this, we observe no evidence of leprosy association in Malawi or Mali at 6 of the 7 non-HLA loci at which we have adequate study power to assess this. Interestingly, while we observe no evidence of association with leprosy overall in Indian-ancestry individuals, we do see an effect of rs2015583 genotype on risk of multibacillary leprosy in this population, with an effect size and direction of effect consistent with that observed in African populations. This observation differs from our findings in African populations, where the effect of rs2015583 is not restricted to multibacillary disease, and the effect of this locus on leprosy risk in India will need to be clarified in future studies.

We hypothesized that some of these inter-population differences may reflect differential effects of genetic risk loci on multibacillary and paucibacillary disease. While we find no evidence to support this in the African data, the observation that rs2015583 appears to have multibacillary disease-specific effects in Indian-ancestry populations is intriguing and suggests that this effect may play a role in inter-population heterogeneity of effect in some settings. In addition, differential linkage disequilibrium between assayed variation and a shared causal locus may explain some of the observed inter-population genetic heterogeneity of leprosy risk. In keeping with this we observe modest evidence of leprosy association at *RAB32*, which is distinct from that reported in Chinese populations. Understanding whether genetic variation at *RAB32* is associated with leprosy risk in African populations, and whether this is distinct from that observed in Chinese populations, will require replication in additional study populations. It is also important to note that Malawi and Mali are themselves distinct populations, and fixed-effects meta-analysis without modelling inter-population ancestry differences, for instance with MR-MEGA [34], may have compromised study power.

Our data provide further support for the observation that leprosy and inflammatory bowel disease have a shared genetic architecture [13, 35]. We expand on this observation, identifying pleiotropy at the leprosy-associated locus with inflammatory bowel disease and childhood-onset asthma, suggesting a model in which selection pressure imposed by *M. leprae* has shaped the evolution of both autoimmune and atopic disease in modern populations. We also identify evidence for recent positive selection at 10q24.32, which overlaps with the leprosy-associated locus, in particular in African populations. The identification of recent positive selection at this locus and of pleiotropy to inflammatory bowel disease risk is consistent with previously-published data demonstrating enrichment for signatures of natural selection among inflammatory bowel disease susceptibility loci identified by GWAS [13]. Selection pressure imposed by leprosy is unlikely to have operated through increased mortality, but may have done so through reduced fertility among cases [36]. Indeed men and women with lepromatous leprosy have been reported to have a 40% and 85% reduction in birth frequency respectively [37]. It is also plausible that there have been additional agents of selection pressure operating at this locus, for instance infections with high mortality in early life, and that risk of leprosy and auto-immune/atopic disease have been jointly shaped by an unidentified, additional selective force.

Integrating data describing chromatin state [16] and eQTL mapping [17–21, 38] in a range of tissues we identify robust evidence of regulatory activity at the leprosy-associated locus. One of the credible set SNPs we identify in the leprosy GWAS, rs2274351:T, is located in an active promoter at the 5' ends of two genes (*ACTR1A* and *SUFU*). In keeping with this, the leprosy association at 10q24.32 colocalises with determinants of gene expression in CD4$^+$ T cells, skin and peripheral nerves. In CD4$^+$ T cells the leprosy-associated locus is a determinant of

*ACTR1A* expression, however in skin and peripheral nerves it determines *TMEM180* expression. To help address this ambiguity, we explored whether expression of *ACTR1A* or *TMEM180* is associated with leprosy disease in previously-published transcriptomic studies [22–25]. While there is no evidence for differential expression of *TMEM180* in the context of leprosy, we find evidence that *ACTR1A* expression is down-regulated in leprosy lesions compared to the skin of healthy controls, and that it is also differentially expressed according to clinical subtype of leprosy. Taken together, chromatin state, eQTL mapping data and differential gene expression in the context of leprosy, identifies *ACTR1A* as a novel candidate mediator of leprosy susceptibility. It is noteworthy, however, that there are limitations to our regulatory mapping in this context. Performing colocalisation between a leprosy GWAS in African-ancestry individuals and eQTL mapping data from individuals of European ancestry makes the assumption that the underlying causal regulatory determinants are shared between populations. While population-specific eQTLs are uncommon [39], they are well described [40]. Similarly, the observation that the leprosy associated signal colocalises with two different genes in different tissues highlights the high levels of regulatory activity at this locus, and it remains plausible that leprosy susceptibility at this locus operates in a tissue or context not assessed here, and that pleiotropy with other traits at this locus could operate through distinct regulatory mechanisms.

Our leading candidate mediator of leprosy risk at 10q24.32, *ACTR1A* expression in CD4$^+$ T cells, is highly biologically plausible as a determinant of leprosy. *ACTR1A* encodes actin-related protein-1 (ARP1), a component of the dynactin complex. Dynactin interacts with the cytoplasmic motor, dynein, facilitating intracellular trafficking of a wide range of intracellular cargos [41, 42]. In T cells specifically, dynactin/dynein complexes direct the accumulation of TCRs and secretory vesicles at the immunological synapse [43, 44], and are required for nuclear translocation of NF$\kappa$B in response to T cell stimulation [45]. An eQTL operating in CD4$^+$ T cells modifying leprosy risk reflects the established role of T cell-mediated immunity in leprosy biology. The spectrum of leprosy disease is defined by host T cell responses, with tuberculoid disease being characterised by robust CD4$^+$ T cell IFN$\gamma$ responses and patients with lepromatous disease failing to mount anti-*M. leprae* cell-mediated responses [46]. Similarly, leprosy reversal reactions, characterised by painful inflammation of leprosy lesions, are associated with infiltration of IFN$\gamma$ producing CD4$^+$ T cells [47]. In addition, the highly reproducible association between MHC class II alleles and leprosy strongly suggests a role for *M. leprae* antigen presentation as a determinant of leprosy susceptibility *per se*. Our data are complementary to these observations, suggesting a model in which CD4$^+$ T cell activation, determined at the level of the T cell as well as that of the antigen presenting cell, modifies susceptibility to leprosy.

Here we define a novel susceptibility locus for leprosy in African populations. Genetic variation at that locus has been subject to recent positive selection pressure, and has pleiotropic effects on hematological indices and risk of inflammatory bowel disease and childhood-onset asthma. Consistent with previous data describing a shared genetic architecture for leprosy and inflammatory bowel disease, our study lends weight to the hypothesis that infections, including leprosy, have directed the evolution of autoimmune and atopic disease. Integrating chromatin state, eQTL mapping and transcriptomic data we identify *ACTR1A* as a candidate effector of leprosy risk at this locus, an observation that has the potential to deepen our understanding of leprosy biology, which will be key in informing the development of novel control strategies. By performing a GWAS for leprosy in Africa we are able to identify trait-associated variation, which has not been identified in previously-published, large-scale studies in populations outside Africa. As such, our study emphasises the utility of defining the genetics of disease risk across multiple populations.

## Materials and methods

### Ethics and consent

Following explanation of the study, cases and controls were recruited following verbal consent of the participant. Children were recruited following written, informed consent of their parent/guardian. The study protocol detailing recruitment and sample collection within KPS, Malawi was approved by the National Health Sciences Research Committee of Malawi and by the Ethics Committee of the London School of Hygiene and Tropical Medicine. The study protocols detailing recruitment and sample collection at the Institut Marchoux, Mali and at the Centre Hospitalier Universitaire Gabriel Toure, Bamako, Mali were approved by the University of Bamako Ethics Review Board. Genotyping and imputation for additional Malian controls were collected as part of the MalariaGEN project, for which the study protocol was reviewed by Oxford University Tropical Research Ethics committee (OXTREC), Oxford, UK (OXTREC 020–006). The overall study design, including re-appraisal of study samples using genome-wide genotyping, was reviewed and approved by Oxford University Tropical Research Ethics committee (OXTREC), Oxford, UK (OXTREC 560–15).

### Study participants

Leprosy case and control samples were recruited to the study in Karonga, Malawi and Bamako, Mali. Participant recruitment in Malawi [48, 49] and Mali [50] have been described previously. In brief, cases of leprosy were diagnosed in both settings on the basis of clinical examination, split skin smear and histopathologic examination of biopsy material. Adults or children with "certain" or "probable" cases were considered eligible for recruitment [51]. Cases were further defined as having paucibacillary or multibacillary disease on the basis of clinical examination and bacteriological index >1 on slit-skin smear or biopsy.

In Malawi, cases (n = 327) and controls (n = 436) were recruited to the study within the KPS, a long-term community-based, epidemiological study in Northern Malawi [52]. Leprosy cases were identified through active population surveys in the 1980s, followed by enhanced passive case detection in the 1990s. Control samples were individuals within the KPS with no history or clinical features of leprosy, matched to case samples with respect to age, sex and geographic area of residence.

In Mali, between April and June 1997, patients with leprosy (n = 247) presenting to Institut Marchoux, Bamako, Mali (formerly Mali's national leprology center now Hôpital Dermatologique de Bamako) were recruited to the study as described previously [50]. Control participants (n = 185), following exclusion of leprosy by clinical examination and history, were recruited in the same study setting among hospital staff and patients with diagnoses other than leprosy. In addition, we supplemented control numbers in the Malian replication study using healthy control samples (n = 183), also recruited in Bamako, collected as part of the MalariaGEN project [53, 54].

Cases in Malawi were recruited at a median age of 47 years (range 15 to 83 years) and 187 were female (57%). Among leprosy cases in Malawi, 47 had multibacillary disease and 275 paucibacillary disease. Control samples in Malawi were recruited at a median age of 44 years (range 15 to 82 years) and 273 were female (63%). Cases in Mali were recruited at a median age of 45 years (range 13 to 85 years) and 136 (55%) were female. Among leprosy cases in Mali, 165 had multibacillary disease and 82 paucibacillary disease. Control samples in Mali were recruited at a median age of 30 years (range 14 to 72 years) and 113 were female (62%). Additional MalariaGEN control samples were recruited at a median age of 3 years (range 0 to 15 years) and 91 (50%) were female.

## Genotyping

Genomic DNA was extracted from study samples as described previously [48, 50]. Following quantification [55], samples were genotyped using the Illumina African Diaspora Power Chip platform [56] and genotypes called using GenCall in GenomeStudio (Illumina). Using consensus strand information from the array manifest file and a remapping pipeline (Dr William Rayner, Wellcome Centre for Human Genetics, Oxford) we aligned genotypes such that all alleles are on the forward strand. Throughout genetic positions reflect GRCh37.

## Sample quality control

We calculated per sample quality control (QC) metrics in PLINK [57]. For each sample we calculated the proportion of missing genotype calls, heterozygosity and the mean X and Y channel intensities. We plotted mean X and Y channel intensities (S8 Fig) and missingness against heterozygosity (S9 Fig), defining outlier samples using ABERRANT [58]. We used PLINK to estimate sample sex from genotype data, excluding samples with discordant genotype and metadata sex information. To identify duplicated and related samples, we calculated pairwise relatedness between samples in PLINK. We considered samples with relatedness $> 0.75$ to be duplicates, and additionally identified samples with relatedness $> 0.2$. In both cases we retained the sample with the highest genotyping call rate of a duplicated/related sample pair. We calculated principal components (PC) in EIGENSTRAT [59]. To identify population outliers, we plotted study sample PCs against a background of African Genome Variation Project [60] samples, identifying outliers by visual inspection (S10 Fig). For both PC and relatedness computations we used an LD-pruned SNP set with regions of high linkage disequilibrium (LD) excluded. The first two major principal components differentiate self-reported ethnicity in both Malawi (S11 Fig) and Mali (S12 Fig).

## SNP quality control

Prior to genome-wide imputation, we extracted genotypes from non-duplicated, autosomal SNPs and applied the following SNP QC filters; SNP missingness $> 10\%$, minor allele frequency (MAF) $< 1\%$, Hardy-Weinberg equilibrium (HWE) $p < 1 \times 10^{-20}$ and plate effect $p < 1 \times 10^{-6}$. HWE was calculated among control samples for each cohort. Plate effect represents an association test of nondifferential missingness with the plate on which each sample was genotyped.

## Imputation

Following sample (S12 Table) and SNP (S13 Table) QC, genotypes at 351,236 autosomal SNPs from 612 samples (Malawi) and genotypes at 367,433 autosomal SNPs from 350 samples (Mali) were taken forward for phasing and genome-wide imputation. We performed phasing of genotypes in SHAPEIT2 [61], phasing genotypes across each chromosome for all samples from each country jointly. We then imputed untyped autosomal genotypes using IMPUTE2 (v2.3.0) [62, 63], in 5Mb chunks using the 1000 Genomes Phase III as a reference panel. We used 250kb buffer regions and effective sample size of 20,000.

## HLA imputation

We used HLA*IMP:03 [64] to impute classical HLA alleles into our datasets. As input to HLA*IMP we used genotypes passing sample and SNP QC thresholds in the HLA region (chr6:28Mb-36Mb). HLA*IMP:03 uses a multi-population reference panel, including

individuals of African ancestry. HLA imputation performed well, with estimated imputation accuracies ranging between 95% and 99.8%. We took forward HLA alleles present in both Malawi and Malawi (MAF > 1%), including 93 classical alleles (42 class I and 51 class II) in downstream analysis.

## Additional cross-platform quality control

We noted that relatively few Mali control samples (n = 142) were available for inclusion in the association analysis. To address this, we used genotypes from additional control samples (n = 183) of representative ethnicity collected as part of the MalariaGEN project [53, 54]. These samples have been genotyped on the Illumina Omni 2.5M platform. Sample genotypes have been phased using SHAPEIT2 [61] and untyped genotypes imputed genome-wide using IMPUTE2 (v2.3.2) [62, 63] with 1000 Genomes Phase III as a reference panel. The SNP and sample QC used in processing these samples [53] is highly analogous to the QC we applied to our study samples. MalariaGEN SNP QC excluded poorly genotyped SNPs using the following metrics; SNP missingness (thresholds 2.5–10% dependent on study population), MAF < 1%, HWE $p < 1 \times 10^{-20}$, plate effect $p < 1 \times 10^{-3}$ and a recall test quantifying changes in genotype following a re-clustering process $p < 1 \times 10^{-6}$. MalariaGEN sample QC excluded samples prior to imputation according to the following metrics; channel intensity, missingness, heterozygosity (outlier thresholds determined by ABERRANT [58]), population outliers and duplicated samples (relatedness > 0.75). Of note, related samples (relatedness > 0.2) are retained for imputation purposes.

We defined a shared subset of SNPs genotyped and passing SNP QC on both platforms ($n = 26, 136$), from which we calculated relatedness estimates and PCs. We removed one of related pairs (relatedness > 0.2) from the MalariaGEN samples, which resulted in a final sample size of 519 (208 cases, 311 controls). Inspection of the PCs demonstrate no further population outliers, and that the 10 major PCs are nondifferential with respect to genotyping array (S12 Fig).

## Association analysis

Following imputation, SNPs were taken forward for association analysis if they passed the following QC metrics; MAF >4%, imputation info score >0.4, HWE $p > 1 \times 10^{-10}$. For the Mali samples, these QC filters were applied overall and for samples genotyped on each genotyping platform individually. At each variant passing QC we tested for association with leprosy case-control status in a logistic regression model in SNPTEST [65] in each cohort. At loci of interest, we used multinomial logistic regression, implemented in SNPTEST, to estimate the effect of the genetic variation on leprosy risk stratified by multibacillary and paucibacillary disease. We used control status as the baseline stratum and cases of multibacillary and paucibacillary leprosy as strata. To account for confounding variation, in particular population structure, we included the six major principal components of genotyping data in all models. In addition, in Mali, we included genotyping platform as an additional categorical covariate. At variants passing QC thresholds in both cohorts, we then performed genome-wide meta-analysis under a frequentist fixed-effects model using BINGWA [66]. For association analysis using HLA allele imputations we coded posterior probabilities of each HLA allele to represent carriage of 0, 1 or 2 copies of that allele. Association analysis and meta-analysis was performed in SNPTEST and BINGWA as above. For HLA association analysis we corrected for the number of classical alleles tested (n = 93) and considered FDR <0.05 to be significant.

## Bayesian comparison of models of association

We compared models of association at loci of interest with multibacillary and paucibacillary leprosy, as estimated by multinomial logistic regression, using a Bayesian approach. We considered four models of effect, defined by the prior distributions on the effect size:

"Null": effect size = 0, i.e. no association with leprosy.

"MB": effect size $N(0, 0.2^2)$ for multibacillary disease, but no effect in paucibacillary disease.

"PB": effect size $N(0, 0.2^2)$ for paucibacillary disease, but no effect in multibacillary disease.

"Both": effect size $N(0, 0.2^2)$ and fixed between multibacillary and paucibacillary disease ($\rho = 1$).

For each model we calculated approximate Bayes factors [67] and posterior probabilities, assuming each model to be equally likely a priori. Statistical analysis was performed in R.

## Definition of credible SNP sets

We used a Bayesian approach to identify a set of SNPs with 99% probability of containing the causal locus at the leprosy susceptibility locus at chr10q24.32. Approximate Bayes' factors [67] were calculated for each SNP in the region (a 200kb surrounding rs2015583) with a prior distribution of $N(0, 0.2^2)$. All SNPs were considered equally likely to be the causal variant a priori. A set of SNPs with 99% probability of containing the causal SNP was defined as the smallest number of SNPs for which the summed posterior probabilities exceed 0.99.

## Association analysis at rs2015583 in non-African populations

To assess the effect of rs2015583 genotype on leprosy risk in Chinese populations, we obtained summary statistics of previously-published GWAS of leprosy susceptibility in 6,316 individuals (2,743 cases, 3,573 controls) [5, 8, 9]. Effect estimates were combined across all Chinese studies and populations in a fixed effects meta-analysis.

We further sought to assess the effect of rs2015583 genotype on leprosy risk in Indian ancestry individuals, leveraging a previously-published sample collection of 258 leprosy cases and 300 healthy control samples genotyped on the Illumina IBC gene-centric 50k array [11]. Sample quality control has been previously described [11], with samples with call rates <90%, pairwise IBD>0.2, outlier heterozygosity and ancestry (as estimated by principal component analysis of genotyping data) being excluded from the association analysis, resulting in a final sample size of 448 individuals. The locus of interest is not directly typed on the Illumina IBC gene-centric 50k array, and we therefore imputed genotypes at rs2015583 in the study samples. To maximise imputation accuracy we imputed genotypes in a 2Mb window centered on rs2015583 using IMPUTE2 (v2.3.0) [62, 63] without prephasing, with 1Mb buffer regions and using 100 template haplotypes for phasing. Prior to imputation we excluded SNPs with SNP missingness >10%, MAF <1%, Hardy-Weinberg equilibrium (HWE) $p < 1 \times 10^{-20}$. Imputed genotypes at rs2015583 passed post-imputation quality control thresholds; MAF = 0.38, imputation info score = 0.43, HWE $p$ = 0.16. We performed association analysis in 448 individuals, comprising 209 leprosy cases (108 multibacillary, 101 paucibacillary) and 239 controls. We tested for association between leprosy case-control status and genotype at rs2015583 in a logistic regression model. We used multinomial logistic regression to estimate the effect of the genotype at rs2015583 on leprosy risk stratified by multibacillary and paucibacillary disease. In both models we included the six major principal components of genotyping data (S13 Fig) to account for confounding variation. Association analysis was performed using SNPTEST [65].

## Association analysis at rs2015583 in tuberculosis

To assess the effect of rs2015583 genotype on tuberculosis risk in Ghana and The Gambia, we analysed data from previously published GWAS of tuberculosis in these populations [68]. Gambian samples (1,316 cases, 1,382 controls) were genotyped on an Affymetrix GeneChip 500K array and Ghanaian samples (1,359 cases, 1,952 controls) on an Affymetrix Genome-Wide Human SNP 6.0 Array. We performed imputation in both sample sets using SHAPEIT2 [61] and IMPUTE2 (v2.3.0) [62, 63] using the 1000 Genomes Project Phase III as a reference panel. Genotypes at rs2015583 were well-imputed, imputation info score = 1.00 and 0.97 in Ghana and The Gambia respectively. The minor allele at rs2015583 was common in Ghana and The Gambia (MAF = 0.33 and 0.43 respectively).

## Signatures of positive selection

To assess evidence for recent positive selection at leprosy associated loci we downloaded iHS estimates in 10kb windows genome-wide in all 1000 Genomes Project Phase III samples from https://pophuman.uab.cat/ [69]. Within each population, we calculated rank p-values for iHS at each genomic region as the proportional rank for the iHS genome-wide. We considered a genomic region with a genome-wide rank $p < 0.05$ to constitute evidence of selection.

## Functional annotation and pleiotropy

We downloaded chromatin state segmentation data across nine cell types (H1 ES, embryonic stem cells; K562, erythrocytic leukemia cells; GM12878, B-lymphoblastoid cells; HepG2, hepatocellular carcinoma cells; HUVEC, umbilical vein endothelial cells; HSMM, skeletal muscle myoblasts; NHLF, normal lung fibroblasts, NHEK, normal epidermal keratinocytes; HMEC, mammary epithelial cells) from: http://genome.ucsc.edu/cgi-bin/hgFileUi?db=hg19&g=wgEncodeBroadHmm. These data integrate data from nine chromatin marks to divide the genomic regions into 15 functional states; active promoters, weak promoters, inactive/poised promoters, strong enhancers (x2), weak/poised enhancers (x2), insulators, transcriptional transition, transcriptional elongation, weakly transcribed, polycomb repressed, heterochromatin/low signal and repetitive/CNV (x2) [16]. We assessed overlap of these functional states within each cell type with the 99% credible SNP set (n = 32) defining the leprosy association at chromosome 10q24.32. At rs2274351 we assessed evidence for overlap with transcription factor binding motifs using RegulomeDB v2.0: https://regulomedb.org/ (accessed 24th May 2022).

We used the R package coloc [70] to identify evidence of causal variants shared by eQTL in primary immune cells and GWAS-identified trait associated loci (including leprosy). Coloc adopts a Bayesian approach to compare evidence for independent or shared association signals for two traits at a given genetic locus. We tested for colocalization between leprosy susceptibility at the chr10q24.32 locus and previously-published *cis* eQTL mapping studies in skin (sun exposed, n = 605; sun unexposed, n = 517) and tibial nerve (n = 532) from Genotype-Tissue Expression (GTEx) Project V8 [38] and in naïve and stimulated primary immune cells from individuals of European ancestry [17–21]; NK cells (n = 245), B cells (n = 283), monocytes (n = 414), CD4+ T cells (n = 293), CD8+ T cells (n = 283), neutrophils (n = 101), LPS-stimulated monocytes (2 hours, n = 261; 24 hours, n = 322) and IFN$\gamma$-stimulated monocytes (n = 267). We downloaded GTEx Analysis V8 summary statistics from https://gtexportal.org/home/ (accessed 05/18/2022). We considered evidence for colocalization for each gene within a 250kb window of the peak leprosy association (rs2015583). We considered a posterior probability >0.8 supporting a shared causal locus to be significant.

To assess evidence for pleiotropy with other disease traits we again used coloc to test for the presence of a shared causal locus between the leprosy association at chromosome 10q24.32 and 55 GWAS traits (S5 Table); hematological indices (n = 26) and immune-mediated diseases (n = 13) from the UK Biobank (http://www.nealelab.is/uk-biobank/, accessed 26th March 2021), immune-mediated diseases from the NHGRI-EBI GWAS Catalog (n = 13, ftp://ftp.ebi.ac.uk/pub/databases/gwas/summary_statistics/, accessed 26th March 2021), and inflammatory bowel disease traits from the International Inflammatory Bowel Disease Genetics Consortium (n = 3, https://www.ibdgenetics.org/downloads.html, accessed 26th March 2021). We considered evidence for colocalization between leprosy an each trait for which there was evidence of association ($p < 1 \times 10^{-5}$) within 250kb of rs2015583.

### Differential gene expression analysis in leprosy

To assess differential expression of candidate mediators of leprosy susceptibility in whole blood we used the Tio-Coma et al RNA-Seq dataset [25]. We downloaded RNA-Seq count data from NCBI Gene Expression Omnibus (GEO, GSE163498), before Trimmed Means of M values (TMM) normalisation in edgeR. We correlated log counts per million (log-CPM) for each gene of interest with sample status (household contacts without leprosy, leprosy patients before diagnosis and leprosy patients after diagnosis) using ANOVA, or Kruskal-Wallis tests if the data was non-normally distributed.

To assess whether ex vivo stimulation of whole blood with M. leprae may reveal leprosy-specific regulatory function at the leprosy-associated locus we used the Manry et al microarray and genotyping data from 51 Vietnamese patients with borderline leprosy [24]. We downloaded quality control filtered, genome-wide genotyping data available with the article, before imputation of rs2015583 using Eagle2 and Minimac4 implemented in the Michigan Imputation Server [71]. Genotypes at rs2015583 were well-imputed ($r^2 = 0.995$) and the minor allele is common (MAF = 0.416). We downloaded non-normalised microarray expression data from GEO (GSE100853) removing probes with detection $p < 0.05$ in less than 3 samples. Expression data was normalised with robust spline normalisation in the R package lumi [72]. We correlated genotype at rs2015583 with each probe mapping to a gene in a 250kb window of rs2015583 using linear regression, including 7 principal components of gene expression data in the unstimulated samples and 8 in the stimulated samples (as used in the original publication).

To investigate differential expression of candidate genes in skin from healthy controls and lesions from leprosy patients we used the Belone et al microarray dataset [22] and the Montoya et al RNA-Seq dataset [23]. For the Belone et al microarray data [22] we downloaded background subtracted and LOWESS normalised expression data from GEO (GSE74481). For the Montoya et al RNA-Seq data [23] we downloaded DESeq2 normalised count data from GEO (GSE125943). We compared gene expression between two groups with t-tests (normally distributed data) or Mann-Whitney tests (non-normally distributed data). Comparison of expression across multiple groups was performed by ANOVA, with subsequent pairwise testing with Tukey's HSD tests.

### Supporting information

**S1 Fig. Quantile-quantile plots of leprosy association.** QQ-plots of leprosy association in Malawi (284 cases, 328 controls, SNPs = 10,511,695), Mali (208 cases, 311 controls, SNPs = 10,514,676) and fixed-effects meta-analysis of both populations (cases = 492, controls = 639, SNPs = 9,616,523.
(TIF)

**S2 Fig. Manhattan plots of leprosy association.** Manhattan plots of leprosy association in discovery (Malawi, 284 cases, 328 controls, SNPs = 10,511,695) and replication (Mali, 208 cases, 311 controls, SNPs = 10,514,676) samples. P-value thresholds are annotated on the Malawi Manhattan plot: dashed line, $p = 5 \times 10^{-8}$ (genome-wide significance); dotted line, $p = 1 \times 10^{-5}$ (threshold for suggestive association). In Malawi, 142 SNPs exceed a significance threshold $p = 1 \times 10^{-5}$ and 1 exceeds $p = 5 \times 10^{-8}$. In Mali 176 SNPs exceed a significance threshold $p = 1 \times 10^{-5}$ and 1 exceeds $p = 5 \times 10^{-8}$.
(TIF)

**S3 Fig. Whole blood expression of *ACTR1A* and *TMEM180* in leprosy cases and healthy controls.** *ACTR1A* and *TMEM180* expression in whole blood of household contacts who do not develop leprosy (HHC, n = 40) and those that do (n = 40); before diagnosis (Leprosy pre-Dx) and at the point of diagnosis (Leprosy post-Dx). Data is taken from the Tio-Coma *et al* RNA-Seq dataset [25] (GEO, GSE163498). Gene expression between groups is compared by ANOVA (*ACTR1A*), or Kruskal-Wallis tests (*TMEM180*) if the data was non-normally distributed.
(TIF)

**S4 Fig. Association between gene expression and genotype at rs2015583 in whole blood.** Effect of the rs2015583:G allele on gene expression at 9 genes in *cis* (within a 500kb window) to rs2015583 in whole blood from leprosy patients (n = 51) with (left) and without (right) stimulation with sonicated *M. leprae*. Data is taken from the Manry *et al* dataset [24] (GEO, GSE100853). Genotype at rs2015583 is correlated with gene expression using linear regression, correcting for 7 principal components of gene expression data in the unstimulated samples and 8 in the stimulated samples. Gene expression is not significantly associated ($p < 0.05$) with rs2015583 genotype for any gene with or without stimulation.
(TIF)

**S5 Fig. HLA Leprosy association conditioned on HLA-DQB1\*04:02.** Association statistics represent a fixed-effects meta-analysis of additive association with disease in Malawi and Mali conditioned on HLA-DQB1\*04:02. SNPs are coloured according to linkage disequilibrium to rs2516438, and genotyped SNPs marked with black plusses. Imputed classical HLA alleles are plotted as diamonds, with significantly associated (FDR <0.05) alleles highlighted in blue.
(TIF)

**S6 Fig. HLA Leprosy association conditioned on HLA-DQB1\*04:02 and HLA-B\*49:01.** Association statistics represent a fixed-effects meta-analysis of additive association with disease in Malawi and Mali conditioned on HLA-DQB1\*04:02 and HLA-B\*49:01. SNPs are coloured according to linkage disequilibrium to rs115312361, and genotyped SNPs marked with black plusses. Imputed classical HLA alleles are plotted as diamonds. No significantly associated (FDR <0.05) alleles remain after conditioning on HLA-DQB1\*04:02 and HLA-B\*49:01.
(TIF)

**S7 Fig. Evidence for leprosy association at the RAB32 locus.** (A) Log-transformed odds ratios and 95% confidence intervals of rs2275606 association (peak association in Chinese GWAS data) with leprosy in Malawi, Mali and China. (B) Log-transformed odds ratios and 95% confidence intervals of rs34271799 association with leprosy in Malawi and Mali. (C) Regional association plot of leprosy association at the RAB32 locus. Association statistics represent a fixed-effects meta-analysis of additive association with disease in Malawi and Mali. SNPs are coloured according to linkage disequilibrium to rs34271799, and genotyped SNPs marked with black plusses.
(TIF)

**S8 Fig. Sample X and Y channel intensities.** (A) Mean X and Y channel intensities for Malawi (top) and Mali (bottom) samples. Outlying samples were identified using ABERRANT and are highlighted (orange).
(TIF)

**S9 Fig. Sample missingness and heterozygosity.** (A) Mean sample genotype missingness plotted against heterozygosity for Malawi (top) and Mali (bottom) samples. Outlying samples were identified using ABERRANT and are highlighted (orange).
(TIF)

**S10 Fig. Population outliers.** Plot of the major two principal components of genome wide genotyping data. Malawi study samples are plotted in orange and Mali study samples in green, against a background of African Genome Variation Project samples (gray). Outliers are highlighted (black rings).
(TIF)

**S11 Fig. Principal components of Malawian genome-wide genotyping data.** Individuals are color-coded according to self-reported ethnicity (top) and case-control status; cases in pink, controls in green (bottom).
(TIF)

**S12 Fig. Principal components of Malian genome-wide genotyping data.** Individuals are color-coded according to self-reported ethnicity (top), case-control status (middle; cases in pink, controls in green), and genotyping platform (bottom; Omni 2.5M in purple, Africa Diaspora Power Chip in gray).
(TIF)

**S13 Fig. Principal components of Indian genotyping data.** Individuals are color-coded according to case-control status (middle; cases in pink, controls in green).
(TIF)

**S1 Table. Leprosy association statistics for suggestive associations in Malawi.** Discovery (Malawi), replication (Mali) and meta-analysis association statistics for loci suggestively associated ($p < 1 \times 10^{-5}$) with leprosy risk in Malawi.
(XLSX)

**S2 Table. Leprosy association statistics for suggestive associations in meta-analysis.** Discovery (Malawi), replication (Mali) and meta-analysis association statistics for loci suggestively associated ($p < 1 \times 10^{-5}$) with leprosy risk in the meta-analysis.
(XLSX)

**S3 Table. 99% credible set SNPs for leprosy association at chromosome 10q24.**
(XLSX)

**S4 Table. Study populations and leprosy association statistics at rs2015583 in the Chinese replication analysis.**
(XLSX)

**S5 Table. Analysis of pleiotropy at the leprosy association at chromosome 10q24.** Colocalisation analysis of 55 GWAS traits with the leprosy association at chromosome 10q24.
(XLSX)

**S6 Table. Selection signatures at chromosome 10q24.** Integrated haplotype scores (iHS) and genome-wide log-rank p-values at chr10:104270877–104280877 in 25 1,000 genomes project populations.
(XLSX)

**S7 Table. Chromatin state at chromosome 10q24.** Chromatin state segmentation data (http://genome.ucsc.edu/cgi-bin/hgFileUi?db=hg19&g=wgEncodeBroadHmm) across 9 cell types (H1 ES, embryonic stem cells; K562, erythrocytic leukemia cells; GM12878, B-lymphoblastoid cells; HepG2, hepatocellular carcinoma cells; HUVEC, umbilical vein endothelial cells; HSMM, skeletal muscle myoblasts; NHLF, normal lung fibroblasts, NHEK, normal epidermal keratinocytes; HMEC, mammary epithelial cells) at 33 99%credible set SNPs for the leprosy association at chromosome 10q24.
(XLSX)

**S8 Table. Regulatory function at the chromosome 10q24 leprosy association.** Evidence for colocalisation between regulatory determinants of gene expression at 25 genes in *cis* (within 250kb) to the leprosy association across primary immune cells, skin and peripheral nerve tissue.
(XLSX)

**S9 Table. Association statistics for imputed classical HLA alleles and leprosy risk in Malawi and Mali.**
(XLSX)

**S10 Table. Replication of previously-published leprosy associated non-HLA SNPs in Malawi and Mali.**
(XLSX)

**S11 Table. Replication of previously-published leprosy associated non-HLA SNPs in Malawi and Mali stratified by multibacillary and paucibacillary disease.**
(XLSX)

**S12 Table. Sample quality control summary in Malawi and Mali.**
(XLSX)

**S13 Table. SNP quality control summary in Malawi and Mali.**
(XLSX)

## Acknowledgments

This publication uses genotyping data from the MalariaGEN consortial project [53] (https://doi.org/10.1038/s41467-019-13480-z). We thank Dr Z Wang and Prof. F Zhang for access to leprosy summary statistics in Chinese populations.

We thank Stuart Mucklow and Giles Warner for their assistance in collecting samples in Mali, and Lifted Sichali and Lorren Mwaungulu for their assistance in collecting the Malawi samples. Mali samples and phenotype data were collected between April and June 1997 at the Institut Marchoux (then Mali's leprology centre) [50]. Phenotype data was collected for the Malawi leprosy cases and controls in the 1980s and 1990s as part of epidemiological surveillance within the Karonga Prevention Study, and samples were collected between February 1997 and April 2001 [48, 49]. We acknowledge that the inclusion of researchers and field workers involved in primary data collection in the authorship of subsequent analyses, for which those individuals had no direct input, is a complex issue. We recognize that without the original field work, none of the subsequent genetic epidemiology studies, including the work presented here, would be possible. However, we are mindful that the inclusion as study authors of individuals without involvement in the secondary analysis, may undermine the authorship of individuals who have made substantial intellectual contributions to such studies. The approach we have taken here is to limit study authorship to individuals contributing to the

current study, but recognize that such an approach also has its limitations. Regardless of the approach taken, transparency of researcher contribution and criteria for authorship is important. We welcome debate from the research community on this issue.

## Author Contributions

**Conceptualization:** Tom Parks, Vivek Naranbhai, Samba Sow, Jörg M. Pönnighaus, Amelia C. Crampin, Paul E. M. Fine, Adrian V. S. Hill.

**Data curation:** Lily Goldblatt, Gavin Band, Kirk A. Rockett, Ousmane B. Toure, Salimata Konate, Sibiri Sissoko, Abdoulaye A. Djimdé, Mahamadou A. Thera, Ogobara K. Doumbo, Sian Floyd, Jörg M. Pönnighaus, David K. Warndorff.

**Formal analysis:** James J. Gilchrist.

**Funding acquisition:** James J. Gilchrist, Tom Parks, Adrian V. S. Hill.

**Investigation:** James J. Gilchrist, Kathryn Auckland, Lily Goldblatt, Vivek Naranbhai, Sian Floyd.

**Methodology:** James J. Gilchrist, Tom Parks, Vivek Naranbhai.

**Project administration:** Kathryn Auckland, Tom Parks, Alexander J. Mentzer, Gavin Band, Kirk A. Rockett, Abdoulaye A. Djimdé, Mahamadou A. Thera, Ogobara K. Doumbo, Samba Sow, Jörg M. Pönnighaus, David K. Warndorff, Amelia C. Crampin, Paul E. M. Fine.

**Resources:** Alexander J. Mentzer, Gavin Band, Kirk A. Rockett, Ousmane B. Toure, Salimata Konate, Sibiri Sissoko, Abdoulaye A. Djimdé, Mahamadou A. Thera, Ogobara K. Doumbo, Samba Sow, Sian Floyd, David K. Warndorff, Amelia C. Crampin, Paul E. M. Fine, Benjamin P. Fairfax.

**Supervision:** Abdoulaye A. Djimdé, Mahamadou A. Thera, Ogobara K. Doumbo, Samba Sow, Amelia C. Crampin, Paul E. M. Fine, Benjamin P. Fairfax, Adrian V. S. Hill.

**Visualization:** James J. Gilchrist.

**Writing – review & editing:** James J. Gilchrist, James J. Gilchrist, Tom Parks, Alexander J. Mentzer, Vivek Naranbhai, Adrian V. S. Hill.

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
