## [Decision Letter · Decision Letter 0]

29 Mar 2022

Dear Dr Gilchrist,

Thank you very much for submitting your manuscript "ACTR1A has pleiotropic effects on risk of leprosy, inflammatory bowel disease and atopy." for consideration at PLOS Pathogens. As with all papers reviewed by the journal, your manuscript was reviewed by members of the editorial board and by several independent reviewers. In light of the reviews (below this email), we would like to invite the resubmission of a significantly-revised version that takes into account the reviewers' comments.

The reviewers raised several questions regarding overlap of the present sample with previous published studies, and replication of results in an Indian leprosy sample. There were also concerns regards the functional validation which need to be addressed and the strength of support for a pleiotropic effect.

We cannot make any decision about publication until we have seen the revised manuscript and your response to the reviewers' comments. Your revised manuscript is also likely to be sent to reviewers for further evaluation.

Sincerely,

Erwin Schurr

Associate Editor

PLOS Pathogens

Alexander Gorbalenya

Section Editor

PLOS Pathogens

Kasturi Haldar

Editor-in-Chief

PLOS Pathogens

orcid.org/0000-0001-5065-158X

Michael Malim

Editor-in-Chief

PLOS Pathogens

orcid.org/0000-0002-7699-2064

The reviewers raised several questions regarding overlap of the present sample with previous published studies, and replication of results in an Indian leprosy sample. There were also concerns regards the functional validation which need to be addressed and the strength of support for a pleiotropic effect.

Reviewer's Responses to Questions

**Part I - Summary**

Reviewer #1: This paper took a look at leprosy susceptibility in African populations by conducting a genome-wide association study, while only one novel leprosy locus was identified and replicated known locus on HLA, LACC1 and SLC29A3 with a nominal significance. This approach is unique and commendable. Generally, the paper is well written, although there are some important points that are not clear that must be addressed.

Reviewer #2: Gilchrist et al performed the first leprosy GWAS in African populations and identified an African-specific leprosy-risk locus on chromosome 10q24.32. Moreover, five loci previously associated with leprosy in Asian populations were validated in Malians and Malawians. The GWAS analysis was well conducted. However, the interpretation of the findings requires further discussion.

Reviewer #3: The paper presents a GWAS analysis in Malawi and Mali screening leprosy patients and controls. Among the genes initially pinpointed, data focus on a novel candidate that is validated using public databanks for gene expression suggesting the association of rs2015583 of ACTR1 as an eQTL and also recovering datasets from case-controls in other diseases such as IBD, Atopy, and an interesting effect where the same SNP was associated, although in IBD in an opposite direction. Later, the authors replicated findings of previously associated genes and SNPs in these African populations. The paper is well written, and the authors have a long contribution to the genetics of infectious diseases with seminal published papers.

**Part II – Major Issues: Key Experiments Required for Acceptance**

Reviewer #1: 1. Better to provide a map which described the sample distribution from African population.

2.Sample information summary and statistics of main results should be listed as tables in the main text.

3.How to explain that most published GWAS results were not replicated in African leprosy patients? If there was sample size limitation, should perform a power calculation analysis to identify the power to discover novel locus or replicate known locus. If there was population heterogeneity issue, should also take a look at the allele frequency difference between populations.

4. Current evidence about potential function for the novel locus is only based on published eQTL mapping studies and not enough for the conclusions. The author should perform experiment to validate the eQTL results.

5.The title for this manuscript is over estimated. More work on either genetics or biology for IBD and atopy should be performed before concluded that ACTR1A has pleiotropic effects.

6. HLA-DRB1*15:01 has been well reported for leprosy susceptibility in several populations. What about this signal in your data? How about the LD between reported HLA 4-digit allele and it?

Reviewer #2: 1. In the Manhattan plot for the Malawi and Mali populations combined (Fig 1), it appears to be a suggestive locus on chromosome 3. However, I could not find SNPs with the same magnitude of meta-analysis p-value for chromosome 3 on Table S1. This needs to be clarified. If the signal is correct, what are the potential genes of interest in the chromosome 3 locus? The signal appears to be stronger than the HLA locus in Africans.

2. Although stigmatized, leprosy is not a deadly disease as tuberculosis was during middle age. What would be the rationale to conclude that leprosy caused selective pressure for atopic and autoimmune diseases in modern populations as implied in the abstract and discussion? For example, if leprosy mortality was high, protective alleles (candidates for autoimmunity risk) could have been positive selected. Since mortality was likely the same as in non-leprosy cases one would expect neutrality. In that case, social and environmental factors may have played an important role. While this discussion falls beyond the current study, the authors need to extensively substantiate their discussion regarding leprosy as the cause of selection or tone down the claim.

3. eQTLs can be cell-specific but they can also differ between populations and regarding their response to stimulus across populations (Nédélec, Y et al, Cell 2016 and Randolph, HE et al, Science 2021). The eQTL effect for ACTR1A in CD4+ T cells was detected in Estonians, while the leprosy GWAS signal was observed in Africans. Although I agree that ACTR1A can be a good candidate gene, one cannot disregard the potential role for other genes at the chr10 locus. This should be addressed as a limitation.

4. Since the identification of ACTR1A as a leprosy candidate gene was based on eQTLs in Europeans, are the leading GWAS SNPs eQTLs for other genes in databases such as GTEx? What about the eQTL effect in other tissues or cell populations? Is ACTR1A differentially expressed in leprosy patients? Are the leading SNPs on chr10 response eQTLs for M. leprae (Manry, J et al, PLoS Gen 2017)? There are multiple publicly available datasets (i.e. GEO) that include gene expression profiles in leprosy affected cases and healthy controls (whole blood, different cell types or leprosy lesions). These sources can be evaluated to strengthen ACTR1A as the best candidate gene at the chr10 locus.

5. Does any of the high LD SNPs in chr10 locus colocalize with transcription factor (TF) binding sites or disrupt TF binding motifs? Can the authors go a step further and identify the link between the genotype and the eQTL effect?

6. The Malawi population has the lowest overall allele frequency for leprosy-risk SNPs in chromosome 10 compared to other populations, including Africans from 1k genomes. Are there any known signs of selection for variants in this chromosomal region?

7. Personal communications should be avoided in the final version of the manuscript. The authors referred to a lack of association of chr10 in and leprosy in the Chinese population. What about these SNPs in the Vietnamese or Indian populations?

Reviewer #3: The group has published data on tuberculosis GWAS studies in Africa and it would be helpful if the authors test ACTR1 also in these populations to evaluate whether the ACTR1 is a major gene involved in mycobacterial infections consequently the dynein-dynactin motor function of the cell.

The Malian and Malawian case-control populations have been studied before. It is important to clarify whether all samples from previous studies were used in this work. The cases numbers are very different from previous publications at least as described in Fitness (2004) and Meisner (2001). So, retrieving information from these previous publications does not help in the understanding distribution of the cases according to some key variables according to clinical form outcome. Thus, the distribution of cases between clinical forms (PB and MB) along with age at diagnosis and gender should be included. African populations generally have a higher proportion of PB patients. Although heterogeneity analysis did not show any differences from the clinical forms, this information would be helpful for readers. Also, information on reactional states (% of ENL and RR) is important. Maybe a supplementary table could be introduced.

Also, in 2010, the same group published a paper (Wong et al NEJM) replicating the findings of the Chinese GWAS and found LACC (formerly C13orf31, rs3764147, Ile254Val) in Mali. So, in this case, it seems that the data is already published. This part should be clarified. A possibility should at the metanalysis declare that this data is already published if this is the case.

In this regard, the same paper (Wong et al, 2010) had genotyped 492 patients in New Delhi and 382 in Kolkata for GWAS replication. It should be very interesting to have validation of the findings of this paper (ACTR1) using these Indian populations.

The validation of ACTR1 and functional analysis could be improved if the authors retrieve from public datasets to better understand how M. leprae regulates ACTR1 expression (and maybe the other gene in the locus) irrespective of the SNP. First, RNASeq and microarray datasets are available in GEO or Array express mostly testing skin biopsies (Montoya et al 2009, Belone et al 2015, Lee et al 2010, among others). In addition, Schwann cells or macrophages infected with live M.leprae have been tested (Masaki et al 2013, Toledo-Pinto 2016, Leal-Calvo et 2021a). Novel studies are also available using single cells RNASeq ou spatial mapping.

Another possibility would be to use whole blood, PBMCs, or skin biopsies from leprosy patients (preferably genotyped) to evaluate gene expression and eQTL of ACTR1 in leprosy.

Also, a dataset for eQTL presenting data from whole-blood cells challenged with sonicated M. leprae (Manry et al, PLoS Genetics, GSE100853) is publicly available and could confirm the impact of this SNP on ACTR1 expression in the leprosy context. Also, other SNPs in the genes nearby could be used.

**Part III – Minor Issues: Editorial and Data Presentation Modifications**

Reviewer #1: NA

Reviewer #2: (No Response)

Reviewer #3: The replication strategy of former associated genes and SNPs should be clarified. How SNPs/genes were selected? The approach to defining the 34 SNPs should be described in the methods. There are genes and SNPs identified in Chinese GWAS, exome, and other studies, but why other GWAS were not considered? Some of the SNPs tested are non-informative with frequencies of 0-0.1% in Africans, such as IL23, CCDC88B, FLG, CARD9, CCDC88B, GIT2, ALDH2, HIF1A, IL12B, TCN2 (Supp Table 6), and they should be excluded. A rapid and previous analysis evaluating the frequencies in Africans should be included as the first step of validation. Data on TLR1 from the same group and relevant genes such as PRKN, LRRK2, other SNPs from NOD2 (702W- rs2066844, rs5743291).

In addition, since rs2015583 is associated with leprosy and IBD in opposite directions, gene expression in other datasets could bring clarity to how ACTR1 levels would function in these diseases. Curiously, most of the genes previously associated with both diseases indicate association in the same direction, especially for NOD2 SNPs.

In supplementary Figure 2, there are some SNPs passing the threshold in Malawian or Malian populations, although probably did not reach significance when the populations were combined. Nevertheless, it should be mentioned at least the legends of the supplementary Figure.

Genomic structure in the PCA (sup Fig 8) also indicates an interesting pattern where populations are clustered, which means that some SNPs can indeed be important in Malawian but not Malian or vice-versa. In this context, it is curious that no signal at 10q24 was detected in these populations separately.

PLOS authors have the option to publish the peer review history of their article (what does this mean?). If published, this will include your full peer review and any attached files.

Reviewer #1: No

Reviewer #2: No

Reviewer #3: No
---

## [Decision Letter · Decision Letter 1]

12 Jul 2022

Dear Dr Gilchrist,

We are pleased to inform you that your manuscript 'Genome-wide association study of leprosy in Malawi and Mali.' has been provisionally accepted for publication in PLOS Pathogens.

Best regards,

Erwin Schurr

Associate Editor

PLOS Pathogens

Alexander Gorbalenya

Section Editor

PLOS Pathogens

Kasturi Haldar

Editor-in-Chief

PLOS Pathogens

orcid.org/0000-0001-5065-158X

Michael Malim

Editor-in-Chief

PLOS Pathogens

orcid.org/0000-0002-7699-2064

Reviewer Comments (if any, and for reference):

Reviewer's Responses to Questions

**Part I - Summary**

Reviewer #1: All my comments have been addressed.

Reviewer #2: I am satisfied with the time and effort put in by the authors to address all reviewer’s questions. The addition of different eQTL datasets, tissue-specific gene expression (including from leprosy patients), functional colocalization, and mechanism of selection strengthened the conclusions of the manuscript indicating ACTR1A and TMEM180 as candidate genes in leprosy pathogenesis.

Reviewer #3: no need for improvements.

**Part II – Major Issues: Key Experiments Required for Acceptance**

Reviewer #1: NA

Reviewer #2: No additional experiments

Reviewer #3: The revised version has been consistently improved and authors present new data from reanalysis of public databanks and previous studies. The data is better presented and discussed. Data show specificities of the association with evidences of replication in different populations and an expanded evidence of the functional association. Although data is not straight forwarded it is clear the role of ACTR1 SNP in leprosy susceptibility among African (and likely African-derived populations). It brings up a novel possible pathway involved in leprosy progression and maybe mostly involved in PB leprosy.

**Part III – Minor Issues: Editorial and Data Presentation Modifications**

Reviewer #1: NA

Reviewer #2: No additional comment

Reviewer #3: (No Response)

PLOS authors have the option to publish the peer review history of their article (what does this mean?). If published, this will include your full peer review and any attached files.

Reviewer #1: No

Reviewer #2: No

Reviewer #3: No

---

## [Editor Report · Acceptance letter]

14 Sep 2022

Dear Dr Gilchrist,

We are delighted to inform you that your manuscript, "Genome-wide association study of leprosy in Malawi and Mali.," has been formally accepted for publication in PLOS Pathogens.

Best regards,

Kasturi Haldar

Editor-in-Chief

PLOS Pathogens

orcid.org/0000-0001-5065-158X

Michael Malim

Editor-in-Chief

PLOS Pathogens

orcid.org/0000-0002-7699-2064